# RETRACTED: Self-Gelling Solid Lipid Nanoparticle Hydrogel Containing Simvastatin as Suitable Wound Dressing: An Investigative Study

**DOI:** 10.3390/gels8010058

**Published:** 2022-01-13

**Authors:** Bhumika Gupta, Garima Sharma, Pratibha Sharma, Simarjot Kaur Sandhu, Indu Pal Kaur

**Affiliations:** University Institute of Pharmaceutical Sciences, Panjab University, Chandigarh 160014, India; bhumika.gupta30@gmail.com (B.G.); garimasharma194@gmail.com (G.S.); pratibhapulastya@gmail.com (P.S.); simarjots@gmail.com (S.K.S.)

**Keywords:** excision wound, controlled release, occlusive, antioxidant activity, liquid crystals

## Abstract

Hydrogels, an advanced interactive system, is finding use as wound dressings, however, they exhibit restricted mechanical properties, macroscopic nature, and may not manage high exudate wounds or incorporate lipophilic actives. In this study, we developed a self-gelling solid lipid nanoparticle (SLNs) dressing to incorporate simvastatin (SIM), a lipophilic, potential wound-healing agent, clinically limited due to poor solubility (0.03 mg/mL) and absorption. The study explores unconventional and novel application of SIM. The idea was to incorporate a significant amount of SIM in a soluble form and release it slowly over a prolonged time. Further, a suitable polymeric surfactant was selected that assigned a self-gelling property to SLNs (SLN-hydrogel) so as to be used as a novel wound dressing. SLNs assign porosity, elasticity, and occlusivity to the dressing to keep the wound area moist. It will also provide better tolerance and sensory properties to the hydrogel. SIM loaded SLN-hydrogel was prepared employing an industry amenable high-pressure homogenization technique. The unique hydrogel dressing was characterized for particle size, zeta potential, Fourier transform infra-red spectroscopy, powder X-ray diffraction, differential scanning calorimetry, rheology, and texture. Significant loading of SIM (10% *w*/*w*) was achieved in spherical nanoparticule hydrogel (0.3 nm (nanoparticles) to 2 µm (gelled-matrix)) that exhibited good spreadability and mechanical properties and slow release up to 72 h. SLN-hydrogel was safe as per the organization for economic co-operation and development (OECD-404) guidelines, with no signs of irritation. Complete healing of excision wound observed in rats within 11 days was 10 times better than marketed povidone-iodine product. The presented work is novel both in terms of classifying a per se SLN-hydrogel and employing SIM. Further, it was established to be a safe, effective, and industry amenable invention.

## 1. Introduction

The complex and dynamic process of wound healing can be effectively managed by creating a favorable environment employing suitable wound-dressing products. It is reported that the global annual cost of wound care products including surgical and chronic wound is expected to increase from $15 billion (2022) to $22 billion (2024) [1]. Even though a wide range of wound dressings are already present in the market, there is scope for great improvement and innovation to address some unmet needs viz. incapability to adjust properties in accordance with changing wound condition; pain and secondary injuries (disturbed new epidermis) due to excessive drying; and loss of efficiency on saturation with exudates. Newer interactive dressings like films and foams are associated with the latter issue of excessive exudate accumulation on site, leading to delayed healing [2].

In contrast to other systems, hydrogels are advanced and interactive and have attracted attention due to (i) their high water content that initiates granulation and epithelialization by promoting fibroblast proliferation and keratinocyte migration; (ii) their soft elastic nature, easy application, and removal; and, (iii) soothing and cooling effect on wounds [3].

However, hydrogel wound dressings may present incompatibility with high exudate wounds and exhibit limited mechanical property and non-adherent and macroscopic nature [2].

Statins are reported to elicit wound healing [4,5] beyond their much-claimed cholesterol-reducing effects. More than one biological mechanism has been attributed to their role in cutaneous tissue regeneration, including pro-lymphangiogenic, pro-angiogenic, antiinflammatory, antifibrotic, antibacterial, and immunomodulatory effects [6]. Since, amongst the various available statins, studies on the role of simvastatin (SIM) in wound healing are more frequent [7], it was selected as a model drug.

Though observed to be an excellent wound-healing agent in vitro [6], clinical translation to wound healing products of SIM will be limited due to its low solubility (0.03 mg/mL) and variable absorption. Further, its lactone moiety is reported to undergo hydrolysis in the acidic (approx. pH 2) and alkaline media [8]. In aqueous surfactant solution and in the presence of an initiator, SIM is reported to be susceptible to oxidation at the diene functional group [9]. To overcome poor solubility, many approaches viz. salt formation, use of surfactant, use of prodrugs, and micronization of SIM have been tried. Solid lipid nanoparticles (SLNs) have gained considerable attention in topical application due to high stability, safety, efficacy, and their occlusive film-forming properties on the skin surface [10]. In addition, SLNs offers advantages like increased solubility, protection of actives against degradation, high payload, and the possibility of actives targeting and controlled release characteristics. 

SLN is also compatible for use on inflamed skin due to the non-skin irritation and non-toxic properties of the lipid matrix, making it a suitable option for wound-healing application. 

Following the evaporation of water from the lipid nano dispersion when applied to the wounds, lipid particles form an adhesive layer occluding the surface. The latter maintains a moist environment in and around the wound, promoting fast and efficient wound healing [11]. However, SLN dispersions are usually aqueous and are free flowing, suggesting the need to incorporate into secondary vehicles like hydrogels for better topical application.

In apropos to overcoming the aforementioned issues with current wound dressing options and harnessing advantages of SLNs and hydrogel, presently we endeavor to develop a self-gelling solid lipid nanoparticle wound dressing. To this end, this system provides a cross-over for combinations of hydrogel and nanomaterials. The proposed system is comprised of solid lipids, surfactants, and water. The novelty/uniqueness of the presently reported dressing lies in the choice of its components and eduction of multifactorial properties and effects. ‘Poloxamer’, one of the key components of presently developed dressing, was employed at a concentration where it imparted a four-in-one effect of an (i) emulsifier for SLN formation, (ii) cosolvent for SIM, (iii) gelling agent, and (iv) precipitation inhibitor (PI). 

The latter effect overcomes the issue of SIM precipitation/crystallization, which is often seen in nano systems with a high load of poorly soluble actives [12]. The suitable viscosity achieved with the presently described system will ensure ease of application, patient compliance, sufficient contact time, and negates the need to add a secondary gelling agent as is proposed for other topical applications employing SLNs.

Further, the SIM-loaded self-gelling SLN hydrogel wound dressing was prepared employing high-pressure homogenization technique, which is a high output method. Also, no organic solvent was deployed, and care was taken to avoid common surfactants like tweens. The latter resulted in SIM degradation in solution forms. The developed SIM-SLN hydrogel wound dressing was characterized extensively and employed topically to establish its effects on wound healing in the excision wound model in rats.

Conclusively, presently we report not only on the lesser explored wound-healing property of SIM but also endeavor to develop a novel self-gelling SLN-hydrogel wound dressing system with significant advantages as discussed above. Reported SIM dressing is a new class of hydrogel dressing where SLNs perse contribute to the gel structure. The system was characterized for physical, chemical, and crystal characteristics and release; and evaluated for safety (biocompatibility) and effectiveness (in terms of hydrophobic active loading and wound healing). Atomic and sub-atomic studies may however be undertaken next to describe, understand, and harness the structural behavior of such systems.

## 2. Results and Discussions

### 2.1. UV Spectrophotometric Analysis of Simvastatin

A standard plot of SIM was prepared in methanol, chloroform:methanol mixture (1:1), and 50% *v*/*v* methanol in phosphate buffer pH 7.4 (Table 1), respectively, in the concentration range of 1–12 µg/mL to obtain linear plots in agreement with Beer–Lambert’s law. Each method was validated successfully. The values of linearity (R^2^; Table 1), accuracy (101.42%, 98.87%, and 101.76%), and precision (0.38, 0.30, and 0.19) were found to be acceptable as per ICH guidelines. Similar values of extinction coefficient (Table 1) indicate no change in absorption properties of SIM in different solvent systems. 

### 2.2. Preparation of Simvastatin Loaded Solid Lipid Nanoparticle (SIM-SLN-) Hydrogel Wound Dressing by Hot High-Pressure Homogenization Method

It was a challenge to incorporate SIM (1% *w*/*w*) into SLNs, due to its practically insoluble nature (aqueous solubility 0.03 mg/mL) and proneness to hydrolysis in the presence of certain aqueous surfactant solutions [13], and without employing any organic solvent. It was decided to employ only the high-pressure homogenization (HPH) technique, considering its amenability to industrial production and ease of scale up from lab. Both hot and cold HPH methods were tried. However, the cold homogenization technique was unsuccessful, resulting in large micron-sized particles (>1000 nm).

For hot HPH production, several different lipids and surfactants were screened at different concentrations. Simvastatin (1% *w*/*w*) could be solubilized in the presently selected combination of melted lipidic phase (80 °C). The latter was emulsified with the hot aqueous phase comprising co-surfactant Poloxamer 188 (6% *w*/*w*) and other secondary agents at high speed (9000 rpm). Initially, Poloxamer 407 was employed but it caused high froth upon stirring, which made it practically impossible to pass the emulsion through HPH. Poloxamer 188 was thus selected subsequently. This composition however resulted in a large particle size >500 nm with entrapment efficiency of only 30%. Large drug crystals were also observed in the formulation under the microscope. Different concentrations of poloxamer 188 and further alteration in the composition of the aqueous phase was tried to achieve desirable characters viz. high loading/encapsulation, stability (no precipitation/crystallization of SIM upon keeping), particle size, and consistency of the wound dressing.

Modified composition and process parameters could successfully increase the entrapment to 60% and no or minimal crystallization of SIM was observed upon cooling of formed o/w nano emulsion to result in SLN formulation. When homogenization cycles were increased from three to seven, particle size decreased to 294 nm. The particle size of the formulation was however found to increase to 2203 nm upon cooling due to subsequent formation of SLN-hydrogel (Figure 1a,b). The latter can be attributed to the gelling induced by the inclusion of poloxamer 188 in the formulation [13]. Moreover, the vital parameters like increased loading and no/minimal crystallization of SIM over storage are also attributable to precipitation inhibition properties of poloxamer 188 [14].

### 2.3. Characterization of SIM-SLNs

#### 2.3.1. Optical Microscopy

Optical microscopy was performed to observe the prepared SIM-SLNs for general morphology (Figure 2a) and to confirm that the free drug does not crystallize out. The observations were made at a magnification of 100×. As observed in Figure 2a, SLNs were spherical in shape. The possibility of variation in size of developed particles is also indicated in Figure 2a. This was further confirmed by the particles size analysis as described in results Section 2.3.4 Moreover, the absence of any crystals in the image indicates that the free, unentrapped drug exists in the solubilized form and does crystallize out.

#### 2.3.2. Transmission Electron Microscopy (TEM)

TEM studies were carried out to confirm the morphology and particle size, and to rule out any probable aggregation in the prepared SIM-SLN hydrogel. The best image amongst all the captured TEM pictures is shown here as Figure 2b. The average particles were observed as small, dark, and spherical in shape, measuring approximately 50 nm. These particles are spotted as being enmeshed within a hydrogel matrix (Figure 2b) formed by poloxamer 188 in the formulation [15].

#### 2.3.3. Field Emission Scanning Electron Microscopy (FESEM)

FESEM provides topographical information with virtually unlimited depth of the field. A FESEM image of the formulation confirms the nano gel network. The SIM-SLNs (size 80–140 nm) were observed to be enclosed within the hydrogel matrix (Figure 2c,d).

#### 2.3.4. Particle Size Analysis

The average particle size of the formulation was 294 nm (Figure 3a) when determined hot and within 24 h of preparation. However, it increased drastically to >1000 nm (Figure 3b), upon keeping (Table 2). The latter was due to the formation of a nanoparticle enmeshed hydrogel system formed by poloxamer 188, as also observed under TEM and FESEM [13,15]. A wide peak (Figure 3a) ranging from approximately 50 nm to 4000 nm indicates the presence of nanoparticles and initiation of gelling process, simultaneously. With the progression of time, when complete gelling was achieved, the obtained peak indicates (Figure 3b) the presence of particles starting from approximately 300 nm to 5000 nm and a few even larger due to the presence of gelled matrix.

#### 2.3.5. Zeta Potential

Zeta potential of SIM-SLNs was found to be −28 mV. Zeta potential >±25 mV indicates the formation of a stable nanoparticulate system with lower chances of aggregation post storage for longer time periods.

#### 2.3.6. Drug Assay (TDC)

TDC of SIM-SLN hydrogel dressing was estimated to be 109.0 ± 4.7% (*n* = 3). A high value (exceeding 100%) of TDC could be due to some loss of water and concentration of the dispersion during the process of preparation of SIM-SLN by hot high-pressure homogenization. Drug loading of SIM-SLN reported by other workers was between 1.5–2 mg/mL [16,17], whereas, in the present study we achieved a loading of 10 mg/g of SLN hydrogel. The achieved high loading of SIM not only represents enhanced solubility but also the improved therapeutic concentration that can be achieved in the small amount of hydrogel and subsequently on the wound area following the application of a hydrogel.

#### 2.3.7. Entrapment Efficiency (EE)

The EE of prepared SIM-SLN hydrogel dressing was determined by dialysis method using methanol as the dialysate. The entrapment efficiency of the prepared SIM-SLN hydrogel was 60% (*n* = 3). Other groups have reported higher EE of even 90%, but this was achieved at a 5–10 times lower drug loading, and the SLNs were prepared by employing organic solvents [16,17]. The latter have toxicity issues. It may further be noted that methanol was used as the dialysate in the study, and the latter may also dissolve out SIM entrapped in the shell of SLNs, thus giving an underestimate of EE.

#### 2.3.8. Fourier Transform Infra Red Spectroscopy (FTIR)

The spectrum of free SIM (Figure 4a) shows IR peaks at 3551, 2961, 1705, 1461, and 1386 cm^−1^ corresponding to free O–H stretch, methyl C–H asymmetric stretch, ester C=O stretch, methylene C–H symmetric bend, and gem-dimethyl C–H bend, respectively. Similarly, the lipid (Figure 4b) showed IR peaks at 3418 and 1733 cm^−1^ corresponding to O–H stretching and C-O stretching, respectively.

In the physical mixture as also for SIM-SLN hydrogel dressing (Figure 4c), peaks corresponding to lipid are mostly present but peaks corresponding to SIM are usually missing, except those corresponding to 1461 and 1386 cm^−1^ presently observed at 1467 and 1384 cm^−1^ in the physical mixture. The peak corresponding to methylene C–H symmetric bend was also retained in the SIM-SLNs. In SIM-SLNs, most of the observed peaks correspond to the lipid, indicating the successful encapsulation of SIM within the lipid matrix. Further, the peak corresponding to 3418 cm^−1^ of lipid shifted to 3442 cm^−1^, indicating O–H stretch, while this peak shifted to a lower frequency at 3408 cm^−1^ in SIM-SLNs.

It was further observed that the OH-peak at 3418 cm^−1^ in the spectrum of lipid (Figure 4d) was sharp, whereas it broadened in the spectrum of SIM-SLN due to the probable bonding of the –OH group. Also, several peaks of the free SIM (marked with a bluebracket in Figure 4a) missing in the spectrum of the SIM-SLN indicate interaction/encapsulation within the lipid.

#### 2.3.9. Powder X-ray Diffraction (PXRD)

The overlaid pattern of PXRD of Poloxamer 188, cosurfactant, and lipid, respectively, is shown in the above Figure 5c. PXRD pattern of SIM (Figure 5a) exhibited sharp peaks at 2θ scattered angles 9.44, 17.32, 17.79, and 19.48, indicating its crystalline nature. However, the PXRD pattern of lyophilized SIM-SLN hydrogel dressing (Figure 5b) shows diffuse peaks, indicating its amorphous nature, except for peaks at 2θ scattered angles of 19.11, 21.31, and 23.16, which correspond to those observed for poloxamer 188. These peaks are, however, distinctly different from the characteristic peaks of SIM and lipid. Hence, we may conclude that SIM is now efficiently encapsulated in the lipid matrix of SLNs, and the latter is sufficiently amorphous to adjust significant SIM within its molecules. Further, the presence of peaks corresponding to poloxamer in the SIM-SLNs indicates that the lipidic core of SIM-SLN hydrogel dressing is covered by a poloxamer shell. The latter will ensure stability against aggregation.

#### 2.3.10. Differential Scanning Calorimetry (DSC)

DSC of free SIM exhibits an endothermic peak at 140.18 °C (Figure 6a) corresponding to its melting point at 135–138 °C. Similarly, the lipid shows a peak at 73.11 °C (melting point 72 °C) (Figure 6d). However, in the physical mixture, the peak shifts to 79.53 ° C, which indicates probable solubilization of SIM into the lipid matrix (Figure 6b). In case of SIM-SLNs, a peak corresponding to lipid is observed at 72.62 °C (Figure 6c), while there is a significant downshift and broadening of peak corresponding to SIM from 140.18 °C to 118.03 °C. The broader peak at lower temperatures indicates amorphous (improved solubility) and nano nature. 

#### 2.3.11. Rheology

Rheology is the study of the flow of matter, primarily in a liquid state, but also as ‘soft solids’ or solids under conditions in which they respond with plastic flow rather than deforming elastically in response to an applied force, which is involved in the mixing and flow of materials, their packing into containers, and their removal from container prior to use. It can affect the physical stability of the product. Variation in shear stress was noted for the developed formulation at different shear rates by changing the speed of rotation of the bob and cup rheometer (Table 3). The rheological profile of the gel was obtained by measuring the shear rate at different shear stress values (Figure 7a). The viscosity of the formulation decreased with an increase in shear rate, suggesting a shear thinning system (Figure 7b). Shear thinning is a desirable property of topical gels as it facilitates ease of application. 

#### 2.3.12. Texture Analysis

TPA (texture profile analysis) defines the mechanical parameters in terms of hardness, adhesiveness, cohesiveness, compressibility, and consistency. The TPA graph and calculated mechanical properties of SIM-SLN hydrogel dressings are presented in Table 4 and Figure 7c,d.

The hardness is defined as the maximum peak force during the first compression cycle. The hardness of SIM-SLN hydrogel dressing, which predicts the ease of application on the skin, was 1668.398 g; the value lies within the acceptable limits for topical gel applications. Adhesiveness is defined as the negative force area for the first compression cycle. It is the work required to overcome the attractive forces between the surface of the sample and the surface of the probe, and it is related to bioadhesion [18]. The adhesiveness value was calculated to be −2290 g·s.

TPA also provides the information about cohesiveness. The latter describes the ratio of the area under the force-time curve produced on the second compression cycle to that produced on the first compression cycle. The high value of cohesiveness provides full structural recovery following gel application. In the present study, the cohesiveness value of SIM-SLN hydrogel dressing was calculated as −1238.77 g which is nominal for topical applications [19]. From the results of TPA experiments, it can be concluded that SIM-SLN hydrogel dressing has suitable mechanical properties for topical application.

#### 2.3.13. Determination of pH

The topical preparation should not be significantly acidic or basic as both extremes can cause irritation to the applied area. The pH of the SIM-SLN hydrogel dressing was found to be 7.0, which lies within the usually recommended range (pH 5–7) for topical preparations, permitting the safe use of the gel on the skin. Moreover, it is reported that the hydrolysis of SIM is very slow in buffered aqueous solution and in buffered aqueous surfactant solutions, provided that the apparent pH of the system is approximately 7.0 [8].

#### 2.3.14. In-Vitro Release

From the obtained graph (Figure 8) of the in vitro release of SIM-SLN hydrogel, it is observed that the release is multiphasic. The blue portion of the graph (upto 8 h) almost matches the pattern for release of free SIM from a similarly prepared carbopol-poloxamer gel. Such a pattern is followed until 40% of the drug is released. The latter corresponds to the unencapsulated drug present in the SIM-SLN dispersion. A korsmeyer-peppas model (R^2^ = 0.987) of release was observed in this phase, indicating a linear correlation between the log percent drug release and log time [20]; the release of an active from a polymeric system is described as M_t_/M_∞_ = Kt^n^, where, M_t_/M_∞_ is a fraction of active released at time t, k is the release rate constant, and n is the release exponent. The purple part (10 to 72 h) of the release data represents a first order release (R^2^ = 1) of SIM from the SIM-SLN hydrogel, indicating a linear correlation between time and percent cumulative drug release (Table 5). The pattern indicates a three times extended release of SIM from SLNs with 100% release at 72 h, while all free SIM is released within 24 h.

### 2.4. Animal Studies

#### 2.4.1. Acute Dermal Irritation Studies as per OECD Guidelines 404 

The score for dermal irritation study is compiled in Table 6. Zero score value clearly demonstrates a non-irritant nature of the developed SLN hydrogel dressing when applied to dermal tissues, and hence is concluded to be safe for topical application. The study was conducted in accordance with OECD guidelines. Figure 9 show the test animal photographed before and after the application of the formulation.

#### 2.4.2. In Vivo Efficacy in Excision Wound Model in Rats

Visual Examination and Wound Area

The in vivo excision wound healing potential of the SIM-SLN hydrogel dressing was assessed in rats for different doses in comparison with the commercial povidone-iodine product (5% *w*/*w*; Cipladine^®^) commonly employed for treatment of cuts and wounds. Wound healing was measured in terms of percentage reduction in wound size, with the size of wound on day 1 taken as 100% for each animal. Results are depicted in Figure 10.

Representative images of the reduction in the excision wound size with the passage of time (at 1, 3, 5, 7, 9 and 11 days) are depicted in Figure 11. The marketed preparation, positive control, and free SIM gel groups demonstrated a delayed healing process as compared to the variously dosed SIM-SLN hydrogel dressing. Among the three different doses of SIM-SLN hydrogel dressing (i.e., 0.5 mg, 1 mg and 3 mg), markedly better healing was observed in the case of the 1 mg dose group.

The wounds of groups treated with blank SLNs, 0.5 mg dose SIM-SLN hydrogel dressing, 1 mg dose SIM-SLN hydrogel dressing, and 3 mg dose SIM-SLN hydrogel dressing were covered with a dehydrated crust or scab at day 5, while it appeared on day 7 in the free SIM-treated group. The scab was almost completely dissolved at day 11 in the 1 mg dose of SIM-SLN hydrogel dressing-treated group, while it was still present in other groups. 

In the positive control group, the scab was only partially formed on day 9, while pus was observed in the wounds of animals treated with marketed preparation by the 7th day, indicating probable infection of the wound. Due to excessive inflammation and abscess, the scab was not formed in the marketed formulation-treated group until day 11. On the other hand, the wound was almost completely healed in the case of animals treated with a 1 mg dose of SIM-SLN hydrogel dressing by day 11.

Almost 100% wound closure, which is 5 and 10 times better at wound healing than untreated and marketed preparation-treated groups, was seen with the 1 mg SIM-SLN hydrogel dressing-treated group. Further, a 0.5 mg dose-treated group showed an effect similar to 1 mg of the free SIM group, indirectly showing a two times better effect than free SIM. 

However, a higher dose of 3 mg SIM-SLN hydrogel dressing failed to show effects better than free SIM. It has been reported that the lower doses of statins exhibit better effects, whereas the high doses may have a negative impact [21]. Blank SLNs also showed significant wound healing (comparable to free SIM). The latter may be attributed to the presence of poloxamer 188 and selected secondary agents. These agents are known for their wound healing [15,22] and antioxidant effects [23], respectively. Furthermore, SLNs being occlusive in nature have good adhesion and film-forming capacity. Both the properties aid in speedy wound healing due to increased hydration and skin integrity [24]. A report from our lab earlier established the potential of SLNs as scaffold for attachment of stem cells and as cell differentiation inducers [25]. All these research outcomes indicate SLNs per se as a wound-healing promoter.

Histological Studies

The histological study of the skin of positive control animals, Figure 12a, shows the absence of an intact epidermis. Disruption of epidermis indicates absence of healing. Visible signs of inflammation in deeper layers i.e., muscles and blood vessels, are also observed along with inflammatory exudates as compared to the naïve group animals, Figure 12e. Hence, we can say that even though wound contraction has been initiated in these animals, healing is not complete.

Ulcerated surface epidermis with adequate granulation tissue and abscess below the muscle layer in the subcutaneous area was observed in the animals treated with marketed preparation, Cipladine^®^, Figure 12b. The presence of abscess was also observed upon visual examination on the 7th day (Figure 11). 

The animals treated with a 1 mg dose of SIM-SLN hydrogel dressing, Figure 12d, showed healed inflammation and an intact surface epidermis. Scar tissue was observed with no signs of inflammation in the deeper region. There was an increase in the collagenous mass at the site of injury, indicating regeneration and repair of the tissue. The progressive changes in the epidermal and dermal region include the keratinization and full-thickness epidermal regeneration, largely covering the entire wound area with no debridement crust on the epidermal surface. Upon visual observations, also nearly 100% wound closure was observed on day 11 (Figure 11).

Surface ulcer with slight inflammation was observed in the animals treated with blank SLN formulation, Figure 12c. Dermis was observed to be normal. Partial healing may be attributed to the wound-healing tendency of the poloxamer present in the blank SLNs [22].

Biochemical Estimations

Lipid Peroxidation: The positive control group showed a significant increase in MDA levels. The marketed preparation and blank SLN formulation treatments did not show any effect. However, all animals treated with SIM showed a significant (*p* < 0.001) lowering of MDA levels so as to match the naïve values (Figure 12f).

Catalase Assay: Catalase levels were significantly decreased in the positive control group, indicating high H_2_O_2_ production such that more of catalase was consumed in the process, effectively lowering its concentration in the skin. Marketed preparation (Cipladine^®^) and free SIM-treated groups did not show any effect while all three doses of SIM-SLNs showed significant reversal (similar to naïve control) of catalase (Figure 12g). Strangely, animals treated with blank SLN gel also showed a similar effect. The latter is attributed to the inclusion of Phospholipon 90H (1.5% *w*/*w*) in the SLN formulation, which is an established antioxidant [23].

## 3. Materials and Methods

### 3.1. Materials

Simvastatin and poloxamer 188 and 407 were a kind gift sample from Sun Pharma, India. Lipid and cosurfactant were obtained from Gattefosse, France ex gratia. Formalin, hydrochloric acid, tris-hydrochloric acid, and methyl paraben were purchased from S.D fine chemicals Ltd., Mumbai, India. Bovine serum albumin, carbopol 934, disodium hydrogen phosphate, sodium hydrogen phosphate, thiobarbituric acid and tri-chloro acetic acid were purchased from central drug house Pvt. Ltd., Delhi, India. Chloroform, polyethylene glycol 400, potassium dihydrogen phosphate, and methanol were purchased from Fisher Scientific Pvt. Ltd., Mumbai, India. Hydrogen peroxide, sodium potassium tartarate, and sodium chloride were purchased from LobaChemie Pvt. Ltd., Mumbai, India. Copper sulphate and sodium hydroxide were purchased from Qualigens Fine Chemicals, Mumbai, India. Diethyl ether was purchased from Sisco Research Laboratories, Mumbai, India. Dialysis membrane (cut off of 12,000–14,000 Da) was purchased from Himedia Laboratories Ltd., Mumbai, India.

### 3.2. Methods

#### 3.2.1. Spectrophotometric Method of Analysis for Simvastatin

##### Standard Plot

Standard plot of SIM was prepared in three different solvent systems viz methanol; chloroform:methanol (1:1); and 50% *v*/*v* methanol in phosphate buffer pH 7.4. Each of these systems were employed during the study for SIM estimation in different experiments. Stock solution (*n* = 6) of SIM (50 µg/mL) was prepared by dissolving 2.5 mg of SIM in 50 mL of the respective solvents. Each stock was diluted serially to obtain concentrations rainging from 1 µg/mL to12 µg/mL. Each concentration was analyzed spectrophotometrically at λ_max_ of 238 nm and 239.7 nm for chloroform:methanol mixture (1:1). The obtained absorbance for each dilution was plotted against respective concentrations. The extinction coefficient, 
E1cm1%
, of the drug was then calculated. The developed analytical method was validated in terms of linearity, accuracy, and precision.

#### 3.2.2. Preparation of SIM-SLN Hydrogel Using Hot High-Pressure Homogenization (HPH) Technique

A combination of lipids was melted at 75–80 °C, followed by dissolution of SIM (1%) in it. The above non-aqueous phase was then added at once to the hot aqueous phase containing poloxamer 188, a cosurfactant, and a cosolvent in water. Both the phases were maintained at 75–80 °C prior to mixing. The latter under stirring at 9000 rpm for 8 min with a high-speed stirrer resulted in a coarse pre-emulsion. The obtained emulsion was further passed through HPH at 1000 bar for seven complete cycles. The formed o/w emulsion was allowed to cool to room temperature to result in SIM-SLN hydrogel.

Note: 0.5% *w*/*w* carbopol 934 was also added to formed SLN hydrogel to form a consistent hydrogel.

#### 3.2.3. Optical Microscopy

The developed SIM SLN hydrogel was sufficiently diluted with triple-distilled water to result in SLN dispersion for better observation.

A thin smear of SIM-SLN was prepared on a clean glass slide for preliminary observation using optical microscope. 

#### 3.2.4. Transmission Electron Microscopy (TEM)

SIM-SLN dispersion was spread on a carbon-coated copper grid and examined under TEM at an accelerated voltage of 80 kV to observe morphological parameters viz. size, shape, and aggregation.

#### 3.2.5. Field Emission Scanning Electron Microscopy (FESEM)

The surface of the developed formulation was examined using FESEM by mounting the diluted sample on metal grids using double-sided adhesive tape under vacuum.

#### 3.2.6. Particle Size Analysis and Zeta Potential

The mean diameter of SIM-SLN hydrogel was measured after appropriate dilution (100 X) with triple-distilled water. Zeta potential of the SIM-SLN hydrogel was measured at 25 °C and 23.2 V/cm of electric field. The SIM-SLN hydrogel was diluted a hundred times with triple-distilled water before measuring its zeta potential.

#### 3.2.7. Drug Assay (TDC)

TDC of SIM-SLN hydrogel was determined by disrupting 1 g of SLN-hydrogel using a suitable quantity of chloroform:methanol (1:1). The obtained clear solution was analyzed spectrophotometrically at λ_max_ of 239.7 nm using chloroform:methanol (1:1) as blank. TDC was calculated using the following equation:
Total Drug Content=Calculated amount of drug/g of SLN dispersionActual amount of drugincorporated/g of SLN dispersion×100


#### 3.2.8. Determination of Entrapment Efficiency (EE)

The EE of the prepared SIM-SLN hydrogel was determined using a dialysis bag method. The required length of membrane was soaked for 12 h in double-distilled water. SIM-SLNs (1 g) hydrogel was added to dialysis bag, tied at both the ends, and immersed in 100 mL methanol, which was stirred using a magnetic stirrer. SLNs were taken out of the bag after 2 h and disrupted with suitable quantity of chloroform:methanol (1:1). The amount of SIM retained in the SLN dispersion was determined spectrophotometrically.

%EE=Amount of entrapped drug in 1 g of SLN dispersionTDC of 1 g SLN dispersion×100


#### 3.2.9. FTIR

The IR spectroscopy was employed to elaborate the stereochemistry of the raw material and the formed SLN hydrogel. IR analysis of the SIM, blank, and the SIM-loaded SLN hydrogel and various excipients used in their preparation were done using Perkin Elmer-Spectrum RX-IFTIR. The obtained FTIR spectra were studied for any significant changes.

#### 3.2.10. PXRD

The crystalline/amorphous nature of SIM-SLN hydrogel was confirmed by X-ray diffraction studies. Samples were exposed to CuKα radiation (45 kV, 40 mA) with scanning from 5° to 50° at 2θ with a step size and scan of 0.017° and 25 s, respectively. SIM, molten, and resolidified mixture of lipids, lyophilised SIM-SLN, molten, and resolidified mixture of lipids containing solubilized SIM were also analyzed.

#### 3.2.11. DSC

The thermal analysis of SIM, lipid, physical mixture, and SIM-SLN hydrogel was conducted to estimate any significant changes between different peaks. Samples were added to conventional pans and heated at 10–250 °C with a scanning speed of 10 °C/min.

#### 3.2.12. Rheology

Rheology of the SIM-SLN hydrogel was determined at ambient temperature with a cup and bob rheometer using a 5 g sample. Measurements were done by varying shear rate (0.1 to 100 s^−1)^. The Herschel–Bulkley model was employed to determine the relationship between shear stress (τ) and shear rate (γ) as:τ − τ_0_ = *k*γ^n^
where, τ is shear stress, τ_0_ is yield value, *k* is consistency index, n is flow index, and γ is shear rate

#### 3.2.13. Texture Analysis

The mechanical properties of SLN-hydrogel wound dressing were evaluated using a software-controlled penetrometer. The SLN-hydrogel was transferred to a universal bottle and kept in the ultrasonic water bath to remove entrapped air for 20 min at 37 °C. The probe (back extrusion cell) was compressed two times into SLN-hydrogel at 2 mm s^−1^. A delay period of 15 s was kept between the two compressions. Various mechanical parameters viz. compressibility, hardness, cohesiveness, and adhesiveness of SLN-hydrogel were estimated.

#### 3.2.14. Determination of pH

The pH of the prepared SIM-SLN hydrogel was determined using pH paper and by comparing the color change with the standard strips, following its dipping in SIM-SLNs.

#### 3.2.15. In-Vitro Release

The release profile of different formulations was studied using Franz-diffusion apparatus. Dialysis membrane (2000–14,000 Da) after soaking in double-distilled water for 12 h was loaded with 0.2 g of SIM-SLNs or a corresponding amount of free SIM dispersed in 1% CMC (0.2 mL) or carbopol-poloxamer188 dispersion (0.2 mL), all containing 2 mg SIM. The dialysate (30 mL), maintained at 37 °C, was stirred throughout the time of experiment. Aliquots (1 mL) were withdrawn at different time intervals for 24 h (free drug dispersions) and 72 h (SIM-SLNs). The dialysate was taken as 50% *v*/*v* methanol in phosphate buffer pH 7.4. Cosolvent 8%, corresponding to that used in the SIM-SLN hydrogel, was added to the dialysate used for in-vitro release of SIM-SLN hydrogel to balance the osmotic pressure on either side of the dialysis membrane.

#### 3.2.16. Acute Dermal Irritation Studies (OECD Guidelines 404)

Albino rabbits (female, 6 months; average weight 1.4–1.7 kg) were employed in the study.

The fur of rabbits covering approximately 10% of total body surface (3 cm^2^) was removed 24 h before the application of test substances. Special care was taken to avoid any abrasion of skin, and only animals with healthy skin were selected.

Gauze containing SIM-SLN hydrogel was applied and fixed with transpore tape on the shaved area below the neck of animals for 4 h. Intimate contact and uniform distribution of SLN-hydrogel on the skin was ensured. A small area adjacent to the shaved skin served as control. After 4 h, the gauze was removed, and the covered skin was examined for any signs as detailed below in Table 7.

The in vivo test was performed initially using one rabbit, and the response was recorded.

Once the absence of any signs of irritation was observed in the initial test, a further two rabbits were prepared in the same way to confirm the negative response.

All rabbits were examined for any signs of erythema and oedema, and the response was graded at 60 min, 24 h, 48 h, and 72 h after the removal of the test patch. Signs of dermal reactions were graded and recorded as per Table 7.

#### 3.2.17. In Vivo Efficacy in Excision Wound Model in Rats

Visual Examination and Wound Area

Adult Wistar rats (200–250 g) bred at Central Animal House facility, Panjab University, Chandigarh, India, were utilized for the study. All the rats had complete access to a rodent diet (Ashirwad Industries, Mohali, India) and water. Before starting the experiment, rats were acquainted to the laboratory conditions. The protocols employing animals were ethically approved by Institutional Animal Ethics Committee (IAEC), Panjab University, Chandigarh, India. An excision wound (6 mm diameter) was created in anesthetized rats using biopsy punch. All the rats were randomly divided into eight different groups (*n* = 6) as listed in Table 8. Animals with no wound and treatment were labeled as Group I. Full thickness wound was created in rats of Group II–VIII. Rats with no treatment (positive control) were considered as Group II; Group III–V rats were treated with SIM-SLN hydrogel at 0.5 mg, 1 mg, and 3 mg dose of SIM, respectively. Blank SLN-hydrogel was applied on rats of Group VI. Free SIM as 1% CMC suspension was applied on Group VII. Group VIII rats received treatment with marketed formulation (Cipladine^®^, Cipla Ltd., Batch No. UZE94). The wound healing was measured in terms of unhealed wound area on each alternate day till the 11th day.

Histopathological Studies

At the end of 11 days of the treatment protocol, the rats were sacrificed ethically by cervical dislocation, and the treated area was excised, rinsed with ice-cold saline, and fixed in 10% formalin. The skin sections previously embedded in paraffin blocks were suitably dyed with hematoxylin and eosin stain before inspection under a high-power light microscope. The skin sections were also evaluated for the presence of scab/thrombus, inflammation, epithelialization, and granulation tissue.

Histopathology of skin samples was carried out at Medicos Centre, S.C.O. 821-22, sector 22-A, Chandigarh-160022, India under the supervision of Dr B.N. Datta (MD, FAMS, FICP, Ex-Prof. of Pathology, PGIMER, Chandigarh)

Biochemical Estimation

Preparation of Skin Homogenate: At the end of the 11th day, the rats were sacrificed by cervical dislocation. Rat skin of the wound area was excised and rinsed in ice-cold phosphate buffer saline (pH 7.4) and weighed. A 10% *w*/*v* tissue homogenate was prepared (Figure 13), which was further used for estimating protein content, lipid peroxidation, and catalase level.

Protein Estimation: Biuret method was used for the determination of proteins [26]. Skin supernatant (100 µL), sodium chloride (2.9 mL), and working biuret reagent (3 mL) were mixed together. Biuret reagent was composed of copper sulphate with a solution of sodium potassium tartrate in NaOH (0.2 N). Potassium iodide was added to this solution, and the volume was made up to 100 mL with NaOH (0.2 N). The mixture was allowed to stand for almost 10 min, and its absorbance was taken at 540 nm. The protein content was obtained after preparing a standard plot using bovine serum albumin.

Lipid Peroxidation: Thiobarbituric acid-reacting substances (TBARS) levels were measured for estimation of lipid peroxidation. To 0.5 mL supernatant, 0.5 mL Tris HCl (pH 7.4) was added and incubated at 37 °C for 2 h. To the above mixture, 1 mL 10% TCA (ice cold) was added. The mixture was centrifuged for 10 min at 1000 rpm. The supernatant was collected. To 1 mL of supernatant, 1 mL of 0.067% TBA was added and kept on boiling water bath for 10 min. Next, it was cooled and 1 mL of DDW was added, and absorbance was measured at 532 nm.

Catalase Assay: Catalase level was taken as a direct measure of the decomposition of hydrogen peroxide [27]. To 50 µL of tissue supernatant, 3 mL H_2_O_2_ (30 mmol/L) in phosphate buffer (50 mm, freshly prepared) was added and any change in absorbance was observed at 240 nm for 120 s at regular intervals of 30 s.

#### 3.2.18. Statistical Analysis

The data were statistically analyzed employing ANOVA using Prism 6.01 GraphPad Software. All the *p*-values considered respectively for different studies are indicated in the footnote of each figure.

## 4. Conclusions

Hydrogels, due to high biocompatibility, hydrophilicity, and three-dimensional (3D) porous structure that matches the extracellular matrix, are extensively explored as wound dressings [28]. However, their popularity is partly marred due to maceration of adjacent skin due to exudate retention, requirement of secondary dressing, poor mechanical properties, and easy dehydration if kept uncovered [29]. Presently it is proposed to manage these issues and harness all the advantages of hydrogel by development of an amalgam of SLNs and hydrogel. The former is popularly known to provide good elasticity and occlusivity when applied to the skin surface, and hence will manage the drying of hydrogel, undertoning of adjacent skin, and spreadability. In addition to the nature of the dressing, the inclusion of a good healing agent with suitable antibacterial property to manage infected wounds is also important. Although antibiotics are the preferred choice, the issue of bacterial resistance with antibiotic therapy is becoming a challenge. Similarly, biotoxicity and long-term retention of inorganic metals as alternate antimicrobials also points towards exploring other newer options [28].

Simvastatin, HMG-Co A reductase inhibitor is reported to exhibit pleiotropic effects such as increased vascular endothelial growth factor (VEGF) production, stimulation of angiogenesis, reduced oxidative stress, and improved endothelial function effects, all of which can positively influence wound healing. Furthermore, it also exhibits antibiofilm activities [30].

However, to elicit physiological effects, there is a need to enhance the bioprofile of SIM because it shows low aqueous solubility and is susceptible to hydrolytic degradation. In this study, we described a novel SLN system where the external aqueous phase of the dispersion per se formed a gel structure comparable to liquid crystals upon cooling. SIM-SLN hydrogel exhibited a high drug loading of 10 mg/g of formulation (1% *w*/*w*), which was challenging in light of the fact that it is poorly soluble and the choice of surfactants for developing SIM-SLN was limited due to its proneness to degrade in aqueous surfactant solutions viz. tweens. SIM-SLN hydrogel was prepared using hot high-pressure homogenization method. The achieved loading was 5–10 times more than those reported [16,17] previously. An entrapment efficiency of 60% was achieved, which, though less than that reported by others, is more in terms of the total amount encapsulated per unit weight or volume. Further achieving this encapsulation without employing any organic solvent is a significant achievement. Encapsulating SIM within the lipid core of SLNs in an aqueous media without crystallization involved a tremendous effort and optimization. XRD, DSC, and FTIR spectra confirm the absence of precipitation of the drug and amorphousness of the lipid phase, indicating effective encapsulation of SIM and formation of lipidic nanoparticles. The morphological change in developed SLNs upon cooling from liquid to semi-solid state after 24 h was investigated in terms of particle size. An increase in particle size from 294 nm to 2 µm and broadening of the peak can be attributed to the formation of poloxamer assemblies [31].

The developed wound dressing was established to be safe in vivo. The ease of application, extrusion from container, and esthetics were confirmed to be satisfactory as indicated by texture analysis and rheological studies. The prepared formulations were evaluated for their wound-healing efficacy using an excision-wound model in rats. It is reported that the low doses of topically applied statins exhibit better effects, while these effects may be reversed at higher doses [21]. Similar was observed presently as the effects produced at 1 mg dose showed faster and better healing than the 3 mg dose. Free SIM dispersed in CMC gel also showed effects, though significantly less than SIM-SLN hydrogel, on wound healing. Since the presently used excision model involves whole skin removal, the advantages obtained with improved penetration due to SLN nature are not so evident; however, the occlusion for maintaining moist conditions in the wound and porosity are a definite advantage. Though histology and rate of wound healing was monitored along with the oxidative stress markers i.e., MDA and catalase, it will be appropriate to conduct more elaborate studies like monitoring various molecular markers of inflammation, healing, angiogenesis, and various genetic markers to more elaborately establish the mechanism of action. Apart from appropriate safety, efficacy, and patient compliance, it will also be worthwhile and of scientific interest to study the structural aspects of present formulation in depth and develop a clearer insight and understanding of its mechanical properties and sub-atomic structure.

## Figures and Tables

**Figure 1 gels-08-00058-f001:** Prepared SIM-SLNs (**a**) in liquid state within 24 h of preparation and (**b**) semi-solid after 24 h.

**Figure 2 gels-08-00058-f002:** SIM-SLNs: (**a**) Optical microscopic image at 100×, (**b**) TEM image, (**c**) FESEM image at 60×, (**d**) FESEM image at 500×, showing spherical nanoparticles.

**Figure 3 gels-08-00058-f003:** Particle size within 24 h (**a**,**b**) at 7 days of preparation.

**Figure 4 gels-08-00058-f004:** FTIR spectrum of (**a**) free simvastatin, (**b**) SIM-SLN dressing, (**c**) lipid + SIM physical mixture, and (**d**) lipid.

**Figure 5 gels-08-00058-f005:** PXRD spectrum of (**a**) free SIM, (**b**) SIM-SLN hydrogel, and (**c**) overlay of various excipients: Poloxamer 188; P90H: co-sufactant; COMP: Lipid.

**Figure 6 gels-08-00058-f006:** DSC of (**a**) free SIM, (**b**) physical lipid-drug mixture, (**c**) SIM-SLN dressing, and (**d**) lipid.

**Figure 7 gels-08-00058-f007:** (**a**) Graph between shear rate and shear stress, (**b**) plot of viscosity versus shear rate for SIM-SLN hydrogel, (**c**) texture profile analysis for various parameters of SIM-SLN dressing, and (**d**) spreadability test for SIM-SLN hydrogel.

**Figure 8 gels-08-00058-f008:** In vitro release profiles of SIM-SLN hydrogel (described in two phases—blue and purple) and free SIM in two different vehicles.

**Figure 9 gels-08-00058-f009:** Rabbit photographed before and after application (72 h) of the SIM-SLN dressing.

**Figure 10 gels-08-00058-f010:** Percent wound healing/closure of all groups on different days of observation (*n* = 6). Percentwound closure for day 1 was taken as 0%. All values are significantly different from each other except those marked similarly (*p* < 0.001) as *, #, $, +,
>, =, “, &, a, b, c, d, @, , %, e, !, ∧ and o.

**Figure 11 gels-08-00058-f011:** Photographs depicting the area of wound in various groups.

**Figure 12 gels-08-00058-f012:** Histology of rat wound skin of various groups: (**a**) positive control (no treatment), (**b**) marketed preparation, (**c**) blank, (**d**) 1 mg dose SIM-SLN hydrogel dressing, and (**e**) naïve control. (**f**) Effect of various treatments on lipid peroxidation in the wound skin, (**g**) effect of various treatments on catalase levels in the wound skin. All groups vary significantly from each other except those marked similarly (*p* < 0.0001) as *, a, # and ∧.

**Figure 13 gels-08-00058-f013:** Schematic representation of tissue preparation for biochemical tests.

**Table 1 gels-08-00058-t001:** Various parameters derived from standard curves plotted in different solvents respectively.

Solvent	λ_max_	E1cm1%	y = mx + c	R^2^	LOD (µg/mL)	LOQ(µg/mL)
Methanol	238 nm	530	0.053x + 0.001	0.999	0.70	2.13
Chloroform:methanol (1:1)	239.7 nm	530	0.053x + 0.011	0.998	0.50	1.51
50% *v*/*v* Methanol in phosphate buffer pH 7.4	238 nm	560	0.056x + 0.001	0.997	0.20	0.59

**Table 2 gels-08-00058-t002:** Particle size and PDI of SIM-SLN hydrogel on different days.

Particle Size (within 24 h)	PDI (within 24 h)	Particle Size after 7 Days	PDI after 7 Days
294 nm	0.337	2203.8 nm	0.679

**Table 3 gels-08-00058-t003:** Various parameters obtained by carrying out the rheological analysis of SIM-SLNs.

Time	Viscosity	Shear Rate	Log Shear Rate	Shear Stress	Log Shear Stress	Torque
[s]	[Pa·s]	[1/s]		[Pa]		[mNm]
3	808	0.98	−0.008	794	2.900	2.43
15	394	1.10	0.041	434	2.637	1.33
30	337	1.23	0.090	416	2.619	1.27
45	293	1.39	0.143	407	2.610	1.25
60	251	1.56	0.193	391	2.592	1.20
75	217	1.75	0.243	379	2.579	1.16
90	201	1.96	0.292	394	2.595	1.20
105	173	2.21	0.344	382	2.582	1.17
120	156	2.48	0.394	387	2.588	1.18
135	138	2.79	0.446	385	2.585	1.18
150	125	3.12	0.494	389	2.590	1.19
165	107	3.51	0.545	377	2.576	1.15
180	97.5	3.94	0.595	384	2.584	1.18
195	88.4	4.43	0.646	392	2.593	1.20
210	79.2	4.97	0.696	394	2.595	1.20
225	70.3	5.59	0.747	393	2.594	1.20
240	61.5	6.29	0.799	387	2.588	1.18
255	53.2	7.06	0.849	376	2.575	1.15
270	47.3	7.92	0.899	375	2.574	1.15
285	39.8	8.91	0.950	355	2.550	1.09
300	34.8	10.00	1.000	348	2.542	1.06

**Table 4 gels-08-00058-t004:** Various parameters obtained for SIM-SLN hydrogel by texture analysis.

Parameter	Value
Firmness	3303.95 g
Consistency	36,547.62 g·s
Cohesiveness	−1238.77 g
Hardness	1668.398 g
Spreadability	1173.456 g·s
Stickiness	−1754.353 g
Adhesiveness	−2290 g·s

**Table 5 gels-08-00058-t005:** Linear correlation coefficients obtained by fitting release data into various kinetic models.

	Zero Order (r^2^)	First Order (r^2^)	Higuchi (r^2^)	Korsmeyer-Peppas (r^2^)	Hixon (r^2^)	Model Selected
Blue portion	0.959	0.951	0.938	0.987	0.954	Korsmeyer-peppas
Purple portion	0.897	1.000	0.915	0.915	0.859	First order

**Table 6 gels-08-00058-t006:** Scores of acute dermal irritation study of SIM-SLN dressing in rabbits.

Tissue Reaction	Rabbit 1	Rabbit 2	Rabbit 3	Score
0 h	1 h	24 h	48 h	72 h	0 h	1 h	24 h	48 h	72 h	0 h	1 h	24 h	48 h	72 h
Erythema	0	0	0	0	0	0	0	0	0	0	0	0	0	0	0	0/60
Oedema	0	0	0	0	0	0	0	0	0	0	0	0	0	0	0	0/60
Total score	0/120

**Table 7 gels-08-00058-t007:** Score of skin reactions in acute dermal irritation/toxicity test.

**Erythema and Eschar Formulation**
No erythema	0
Very slight erythema (barely perceptible)	1
Well defined erythema	2
Moderate to severe erythema	3
Severe erythema (beef redness) to eschar formation preventing grading of erythema	4
Maximum possible score: 4
**Oedema Formation**
No oedema	0
Very slight oedema (barely perceptible)	1
Slight oedema (edges of the area well defined by definite raising)	2
Moderate oedema (raised approximately 1 mm)	3
Severe oedema (raised more than 1 mm and extending beyond area of exposure)	4
Maximum possible score: 4

**Table 8 gels-08-00058-t008:** List of groups and treatments received by rats of each group.

Group No.	Treatment Received
Group I	Naïve
Group II	Positive Control
Group III	SIM-SLN hydrogel formulation (dose 0.5 mg)
Group IV	SIM-SLN hydrogel formulation (dose 1 mg)
Group V	SIM-SLN hydrogel formulation (dose 3 mg)
Group VI	Blank hydrogel formulation
Group VII	Free SIM suspended in 1% CMC (dose 1 mg)
Group VIII	Marketed preparation (Cipladine^®^)

## Data Availability

The submitted data forms a part of master’s thesis and has not been published elsewhere.

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
