# Peer review of "Self-Gelling Solid Lipid Nanoparticle Hydrogel Containing Simvastatin as Suitable Wound Dressing: An Investigative Study"

_gels, 2022, doi:10.3390/gels8010058_

Round 1

Reviewer 1 Report

The investigations presented in this study sound good and may be published, but after the modifications based on the following comments:

  1. The entire manuscript could benefit from English language editing for improvement of the writing style and correction of grammatical errors.
  2. I suggest author to write and separate the "Introduction" section into 2-3 paragraphs, in which it will really help people to read through.
  3. At the end of the introduction section the research question should be clearly defined and the hypothesis as well.
  4. The limitations of the study should be mentioned as well as future recommendations.
  5. The discussion and the conclusions are enough speculative. Please, review the section and comment with more criticism the pro and cons of your results, with ideas for the further future assessments

Author Response

Authors reply to the Review Report (Reviewer 1)

The investigations presented in this study sound good and may be published, but after the modifications based on the following comments:

Reply: Thank you for your kind consideration and suggestions. We have now modified the entire manuscript as per your recommendations.

  1. The entire manuscript could benefit from English language editing for improvement of the writing style and correction of grammatical errors.

Reply: all the grammatical errors and spelling errors have been removed from the revised manuscript.

  1. I suggest author to write and separate the "Introduction" section into 2-3 paragraphs, in which it will really help people to read through.

Reply: the introduction is divided into different paragraphs and hope to be more understandable and presentable now.

  1. At the end of the introduction section the research question should be clearly defined and the hypothesis as well.

Reply: one more paragraph has been added at the end of the introduction as suggested.

  1. The limitations of the study should be mentioned as well as future recommendations.

Reply: changes done as suggested in the last paragraph of conclusions.

  1. The discussion and the conclusions are enough speculative. Please, review the section and comment with more criticism the pro and cons of your results, with ideas for the further future assessments

Reply: discussions and conclusions are modified as suggested.

Reviewer 2 Report

Overall, the manuscript is well-presented. The authors of this manuscript focus on the development of self-gelling solid lipid nanoparticles hydrogel for wound dressing application, which was characterised for physico-chemical, morphology, and mechanical properties and in vitro drug assays. The safety of hydrogel was evaluated in an irritation rabbit model. The efficacy evident of wound healing capacity was determined in a rat model of excision wound.

However, some suggestions and comments are highlighted to improve the quality of the manuscript. Please make revisions to the manuscript.

Comments

Title:

As you defined SLN as solid lipid nanoparticles in the introduction, would you prefer to revise nanoparticulate to nanoparticle? Please use the same terminology throughout the manuscript.

Please revise ‘self gelling’ to self-gelling.

Abstract:

Please spell out DSC, PXRD, FTIR, OECD in the abstract.

Please add critical conclusions of the findings.

Introduction:

Can you add the hypothesis of the study in the last paragraph of the introduction?

Methods:

Line 553, 560, 568, 570, 573, 579, 585, 587 - Avoid bullet point. You can write the methodology for animal studies in complete sentences in the paragraph to describe each procedure.

Line 563 – please revise the writing error ‘animalsfor4’.

Table 7 – please add heading title, i.e., ‘score’ for score column.

Scheme 1 – please revise ‘scheme’ to figure. If you can draw the schematic, that would be best.

Please add description of statistical analysis.

Results:

Line 137 – please use the same short form throughout the manuscript. Revise Sim-Sln to SIM-SLN.

Please give more explanation for the result of optical microscopy of SIM-SLNs. You can explain the result in terms of non-crystalized features, particle morphology, and distribution. In that image, we can see loads of features of different sizes. They can be particles or artifacts, or SIM. Please mark in the arrow what you are trying to show here.

For the TEM result, was the measurement of particle size was calculated from the average particle sizes or only from one image? Please clarify this in the result section.

In methods, you mentioned extinction co-efficient. Please add this finding in the result section.

Please add a graph for data TDC and EE in the figure.

The descriptions of DSC, TPA, and in vitro release results should come first before their figures. Please revise.

Figures:

You have too many figures. Please reduce the number of figures. For example, you can combine a few results of hydrogel characterization (i.e., figure 1-4) and make them as one figure with multiple panels. Please revise all figures accordingly.

Please indicate sample size (n) and data as mean with standard error mean or deviation for all findings. Please revise all quantitative figures. 

Figure 1 – you can remove the words figure 1a and figure 1b, instead use panel ‘a’ and ‘b’ as you already write down ‘figure’ in the legend. (this comment is applied to other figures as well). Also, please use the clear photograph for figure 1b as the quality of the image is low.

Figure 2 - Please add a scale bar in the image and briefly explain the results of optical microscopy in the legend.

Figure 3 – The quality image of TEM is very poor. Please provide a higher resolution of the TEM image.

Figure 4 – again, remove the word ‘figure' in the images; instead, use panels ‘a’ and ‘b’ in the images.

Figure 5 – you can remove raw data in figure 5 as you have already mentioned the average particle size in table 2.

Figure 6 – Please remove raw data in figure 6. You can add the data together in table 2.

Figure 7 – Data presentation of FTIR spectrum is very poor. Please superimpose all panels a, b, c, and d with the appropriate scale and make it as one figure. Please mark which peaks are altered in the arrow. 

Figure 8 – Similar comments as figure 7. Also, combine figure 8c with figure 8a-b. Only one legend for one figure.

Figure 9 – Please remove raw data of DSC and re-plot the data as a graph.

Consider combining figures 7, 8, and 9 as one figure with multiple panels.

Figure 10 – Please combine figure 10a and figure 10b as one figure.

Figure 11 – Please combine figure 11a and figure 11b as one figure.

Figure 12 – You did not indicate the error bar for the release profile. Please revise.

Consider combining figures 10, 11, and 12 as one figure with multiple panels.

Figure 13 – Please add a scale bar (ruler size) whenever you take gross images of skin. So, the macroscopic view of the region of interest could be measured.

Figure 14 – Please define each symbol (*#$ etc.) as indicated in the figure.

Figure 15 – Please remove unnecessary images. You can only present one representative image of each experimental group. 

Consider combining figures 13, 14, and 15 as one figure with multiple panels.

Figure 16 – Again, please remove the word ‘figure’ in the images; instead, indicate them with panel 'a-e'. Please present images with a similar magnification view, indicate which region of interest in the histology tissue with the arrow, and please add the scale bars.

Figure 17-18 – Move the statistical description (line 416) to the legend (line 417).

Consider combining Figures 17 and 18 as one figure with multiple panels.
